



# Different strategies to retrieve aerosol properties at night-time with GRASP algorithm

Jose Antonio Benavent-Oltra[1,2], Roberto Román[3], Juan Andrés Casquero-Vera[1,2], Daniel Pérez-Ramírez[1,2], Hassan Lyamani[1,2], Pablo Ortiz-Amezcua[1,2], Andrés Esteban Bedoya-Velásquez[2,4], Gregori de Arruda Moreira[2,5], África Barreto[3,6,7], Anton Lopatin[8], David Fuertes[8], Milagros Herrera[9], Benjamin Torres[9], Oleg Dubovik[9], Juan Luis Guerrero-Rascado[1,2], Philippe Goloub[9], Francisco Jose Olmo-Reyes[1,2] and Lucas Alados-Arboledas[1,2]

[1]Department of Applied Physics, University of Granada. 18071, Granada, Spain.
[2]Andalusian Institute for Earth System Research, IISTA-CEAMA, Granada, Spain
[3]Grupo de Óptica Atmosférica (GOA), Universidad de Valladolid, Valladolid, Spain
[4]Sciences Faculty, Department of Physics, Universidad Nacional de Colombia, Medellín, Colombia
[5]Institute of Research and Nuclear Energy (IPEN), São Paulo, Brazil
[6]Cimel Electronique, Paris, France
[7]Izaña Atmospheric Research Center, Meteorological State Agency of Spain (AEMET), Spain
[8]GRASP-SAS, Remote Sensing Developments, Université de Lille, Villeneuve D'ASCQ, 59655, France
[9]Laboratoire d'Optique Atmosphérique (LOA), UMR8518 CNRS, Université de Lille, Villeneuve D'ASCQ, 59655, France

*Correspondence to*: Jose Antonio Benavent-Oltra (jbenavent@ugr.es)

*Abstract*. This study evaluates the potential of GRASP algorithm (Generalized Retrieval of Aerosol and Surface Properties) to retrieve continuous day-to-night aerosol properties, both column-integrated and vertically-resolved. The study is focused on the evaluation of GRASP retrievals during an intense Saharan dust event that occurred during the Sierra Nevada Lidar aerOsol Profiling Experiment I (SLOPE I) field campaign. For daytime aerosol retrievals, we combined the measurements of the lidar ground-based from EARLINET (European Aerosol Research Lidar Network) station and sun/sky photometer from AERONET (Aerosol Robotic Network), both instruments co-located in Granada (Spain). However, for night-time retrievals three different combinations of active and passive remote sensing measurements are proposed. The first scheme (N0) uses lidar night-time measurements in combination with the interpolation of sun/sky daytime measurements. The other two schemes combine lidar night-time measurements with night-time aerosol optical depth obtained by lunar photometry either using intensive properties of the aerosol retrieved during sun/sky daytime measurements (N1) or using the moon aureole radiance obtained by sky camera images (N2).

Evaluations of the columnar aerosol properties retrieved by GRASP are done versus standard AERONET retrievals. The coherence of day-to-night evolutions of the different aerosol properties retrieved by GRASP is also studied. The extinction coefficient vertical profiles retrieved by GRASP are compared with the profiles calculated by Raman technique at



night-time with differences below 30% for all schemes at 355, 532 and 1064 nm. Finally, the volume concentration and scattering coefficient retrieved by GRASP at 2500 m a.s.l. are evaluated by in-situ measurements at this height at Sierra Nevada Station. The differences between GRASP and in-situ measurements are similar for the different schemes, with

differences below 30% for both volume concentration and scattering coefficient. In general, for the scattering coefficient, the GRASP N0 and N1 show better results than the GRASP N2 schemes, while for volume concentration, GRASP N2 shows the lowest differences against in-situ measurements, around 10%, for high the aerosol optical depth values.

**Keywords:** GRASP, aerosol properties night-time, EARLINET, lidar, lunar

photometer, sky camera, AERONET, SLOPE campaign.

## 1. Introduction

Knowledge of the atmospheric aerosol optical and microphysical properties is important due to their different effects on the Earth-atmosphere radiative budget (IPCC, 2013). The aerosol particles can scatter and absorb solar and terrestrial radiation. Earth-Atmosphere radiative

forcing sign (warming or cooling) is sensitive to aerosol optical and microphysical properties and their vertical distribution (e.g. Boucher et al., 2013). In addition, aerosol particles can act as cloud condensation and ice nuclei and, thus, can modify the development, microphysical properties and lifetime of clouds (e.g. Andreae et al 2004; Boucher et al., 2013). Recent developments in remote sensing have allowed advancing in understanding aerosol globally,

but the characteristic of each system do not allow either a complete characterization of day-to-night, especially in aerosol microphysical properties (e.g. Perez-Ramirez et al., 2012). Understanding day-to-night aerosol properties from remote sensing measurements is essential to advances in aerosol dynamics and changes, which eventually will serve for advancing in aerosol impact on air-quality and climate. Therefore, current efforts are in integrating different

measurements that require advancing in the development of retrieval techniques.

During the last two decades, global and regional networks have been established to get a comprehensive, quantitative, and statistically significant database of atmospheric aerosols. The Aerosol Robotic Network (AERONET; Holben et al., 1998) and East Asian SKYNET (Nakajima et al., 2007) use sun/sky photometer to provide aerosol column-integrated properties

with high temporal resolution. These networks use retrieval techniques that allow the characterization of aerosol microphysical properties (e.g. Nakajima et al., 1996; Dubovik and King, 2000). These networks were focused on daytime measurements but nowadays they are



trying to add night-time aerosol measurements derived from lunar photometry. The developments in moon (Berkoff et al., 2011; Barreto et al., 2013, 2016) and star photometry (e.g. Pérez-Ramírez et al., 2011, 2012; Baibakov et al., 2015) allow the acquisition of night-time measurements, however, these measurements are limited in the inversion algorithms to
retrieve the aerosol microphysical properties (Pérez-Ramírez et al., 2015; Torres et al., 2017).

Lidar networks such as EARLINET (European Aerosol Research LIdar NETwork; Pappalardo et al., 2014), LALINET (Latin American LIdar NETwork; Guerrero-Rascado et al., 2016; Antuña-Marrero et al., 2017) and MPLNET (Welton et al., 2002) provide information about aerosol vertical distribution. However, many of the lidar systems operating in these
networks are basic lidar systems which only have information on the backscatter elastic signals and only allows the retrieval of the vertical profiles of the aerosol backscatter coefficient ($\beta$) by the Klett-Fernald method (Fernald et al., 1972; Fernald, 1984; Klett, 1981, 1985) and corresponding aerosol extinction ($\alpha$) coefficient by assuming constant aerosol extinction-to-backscattering ratio, which is so-called lidar ratio (LR). On the other hand, more advanced lidar
systems implement either Raman (e.g. Ansmann et al., 1992; Whiteman et al., 1992) technique for independent retrievals of aerosol backscatter and extinction measurements. These multiwavelength lidar measurements allow use different inversion algorithms based on the regularization technique to retrieve vertical profiles of aerosol microphysical properties using $3\beta+2\alpha$ configuration; that is multiwavelength lidar measurements of three backscatter and two
extinction coefficients (e.g. Müller et al., 1999; Böckmann et al., 2001; Veselovskii et al., 2002). Nevertheless, the amount advanced lidar systems is considerably lower when compared with basic lidar systems, therefore the independent $\alpha$ and $\beta$ measurements are sparse and mostly limited to night-time. In this context there is a lot of passive and active remote sensing measurements that alone do not allow enough information to retrieving advanced aerosol
microphysical properties. However, integrating all these measurements in an appropriate inversion scheme allows such retrievals and can even complete the number of unknown aerosol optical properties. Such integration is critical for retrieving vertical profiles where the information content for the retrievals is considerable low when comparing with classical sun photometer inversion (e.g. Veselovskii 2005). In the framework of EARLINET different
inversion algorithm were developed, such as the LIdar-Radiometer Inversion Code (LIRIC; Chaikovsky et al., 2008, 2016) that uses as input AERONET retrievals and backscatter elastic signals, and the Generalized Aerosol Retrieval from Radiometer and Lidar Combined





(GARRLiC; Lopatin et al., 2013) code which uses as inputs sun/sky radiance and backscatter lidar measurements that make the inversion more consistent (Lopatin et al., 2013).

Among these algorithms, we use in this study the recently developed Generalized Retrieval of Aerosol and Surface Properties algorithm (GRASP; Dubovik et al., 2014) which includes GARRLiC code. GRASP is a versatile and open-source algorithm (www.grasp-open.com) based in the concept of Dubovik and King (2000) algorithm which has been successfully used by AERONET during the last decades. GRASP algorithm is divided in two main independent modules: forward model and numerical inversion modules. The forward model is based on radiative transfer and aerosol models and it is a convenient tool for sensitivity and tuning studies (Dubovik et al., 2014, Torres et al., 2017). The numerical inversion module is the main part of the core program, which includes general mathematical operations based on multi-term least square method (LSM) concept (Dubovik and King, 2000; Dubovik, 2004). The GRASP versatility allows the retrieval of aerosol properties through the combination of measurements from different instruments both column-integrated and vertically resolved. In fact, GRASP was successfully utilized for the retrieval of the aerosol properties using different configurations and measurements, such as: polar nephelometer data (Espinosa et al., 2017); satellite images (Kokhanovsky et al., 2015); aerosol optical depth (AOD) and sky radiances (including polarization) (Fedarenka et al., 2016); spectral AOD and sky camera images (Román et al., 2017a); only spectral AOD (Torres et al., 2017); and the combination of aerosol optical depth (AOD), sky radiances and elastic lidar (Lopatin et al., 2013; Benavent-Oltra et., 2017) or ceilometer profiles (Román et al., 2018). The aerosol properties retrieved by GRASP aerosol profiles have been used as input to radiative transfer models (Granados-Muñoz et al., 2019), to evaluate dust forecast models (Tsekeri et al., 2017) or to be assimilated in global models (Chen et al., 2018).

In this framework, the main objective of this paper is to propose and explore different and novel strategies for the retrieval of vertically-resolved aerosol properties at night-time using GRASP algorithm combining remote sensing measurements as input data. Another goal is to quantify the accuracy of the retrieved night-time aerosol properties obtained by these strategies, classified in three schemes, using as reference independent aerosol measurements and products. To that end, the recent developments on lunar photometry which allows to derive the night-time AOD from lunar photometer (Barreto et al., 2013, 2016) and the new studies with sky camera images that allow to obtain the normalized sky radiance from lunar aureole (Román et al., 2017a), open the possibility to explore the use GRASP algorithm combining



these night-time measurements with elastic lidar data to study night-time microphysical and optical aerosol properties.

The paper structure is as follows: Section 2 and 3 give a brief description of the experimental site, instrumentation used and the dust event occurred during Sierra Nevada Lidar aerOsol Profiling Experiment I (SLOPE I) campaign. The different schemes used in GRASP to retrieve the aerosol properties both day and night-time are described in Section 4. In section 5, the assessment of the aerosol column-integrated and vertically-resolved properties both day and night-time retrieved by GRASP is discussed. Finally, the conclusions are given in Section 6.

## 2. Experimental site and instrumentation

### 2.1. AGORA Observatory

The paper is mainly focus on the city of Granada (Spain). Granada is located in the Western Mediterranean basin and it is frequently affected by long-range transport of Saharan dust (Lyamani et al., 2005; Fernández et al., 2019; Soupiona et al., 2019) and biomass-burning, both from near sources (Alados-Arboledas et al., 2011) and large distances (e.g. Ortiz-Amezcua et al., 2017; Sicard et al., 2019). Main local sources of anthropogenic aerosols are road-traffic and heating systems during the winter season (Lyamani et al., 2010). Under strong anticyclone conditions, the orographic situation with the city situated in a basin surrounded by mountains makes ventilation processes difficult and favors aerosol stagnation (Patrón et al., 2017).

The experimental measurements used in this study were collected in the AGORA observatory (Andalusian Global ObseRvatory of the Atmosphere) in Granada. AGORA deployed instrumentation at three different stations at different altitudes. The principal station (UGR station) is located in the Andalusian Institute for Earth System Research/IISTA-CEAMA in Granada city where active and passive remote sensing instrumentation operated. The other two stations are in Sierra Nevada Mountain range: Cerro Poyos station (37.11° N, 3.49° W, 1820 m a.s.l.) and Sierra Nevada station (SNS; 37.10° N, 3.39° W, 2500 m a.s.l.). In this study, we used the in-situ measurements from SNS station which is located about 25 km away (horizontally) from UGR station. The measurements of SNS station can allow for characterization of regional and long-range transport episodes and the validation of inversion algorithms used to retrieve aerosol optical and microphysical properties. The altitude difference between UGR and SNS stations (~1.8 km) and the short horizontal distance make that the



correlative measurements between both sites ideal in our objective of evaluating different GRASP scheme retrievals.

The measurements used in this work were acquired in the framework of the Sierra Nevada Lidar aerOsol Profiling Experiment I (SLOPE I) campaign. SLOPE I took place at AGORA from May to September 2016 with the objective to validate the vertically-resolved aerosol properties retrieved from the combination of active and passive remote sensing measurements by in-situ measurements on the surrounding high mountain area. In this regard, several studies have been done using SLOPE I database: day-to-night evolution of planetary boundary layer (de Arruda Moreira et al., 2018) and its turbulence behavior (de Arruda Moreira et al., 2019), aerosol hygroscopic growth (Bedoya-Velásquez et al., 2018), evaluation of the aerosol properties during daytime retrieved by GRASP combining a ceilometer and sun/sky photometer measurements (Román et al., 2018) and the characterization of the angular scattering of the Sahara dust aerosol by means of polar nephelometry (Horvath et al, 2018). Thus, SLOPE I is ideal for our purposes of studying day-to-night aerosol microphysical properties retrievals.

## 2.2. Remote sensing measurements

The measurements of the remote sensing instrumentation of UGR station are used as input data in the different GRASP schemes (see Section 4). One of these instruments is a multiwavelength Raman lidar (LR331D400, Raymetrics S.A.) which is included in EARLINET network since 2005 and contributes to the ACTRIS research infrastructure. It is composed of a pulsed Nd:YAG laser that emits at 1064 nm (110 mJ/pulse), 532 nm (65 mJ/pulse) and 355 nm (60 mJ/pulse) by means of the 2nd and 3rd harmonic generators. The receiving system has seven channels: three to measure the backscatter signal at emission wavelengths plus one additional channel to measure the cross-polarized signal at 532 nm; two channels at 387 and 607 nm for the detection of Raman scattering from $N_2$ and an additional channel to detect the Raman scattering from water vapour at 408 nm. Due to incomplete overlap, atmospheric information up to 500 m above the system is limited (Navas-Guzmán et al., 2011). A detailed description of this multiwavelength Raman lidar system can be found in Guerrero-Rascado et al. (2008, 2009).

Co-located with the lidar system, a sun/sky/lunar photometer Cimel CE318-T (Cimel Electronique), included in the AERONET network, makes day and night-time measurements since March 2016. This photometer is equipped with a filter wheel (9 narrow filters) covering



the spectral range between 340 and 1640 nm. During daytime, the sun/sky/lunar photometer performs measurements of sky radiance but also direct solar irradiance, which is used to derive the AOD; both kind of measurements can be used to retrieve detailed aerosol properties such as particle size distribution, complex refractive index (CRI) and single-scattering albedo (SSA)

(Nakajima et al., 1996; Dubovik et al., 2006). This photometer is annually calibrated following the AERONET methodology by ACTRIS/AERONET-Europe, the European branch of AERONET. Furthermore, this photometer has the capacity to measure the solar radiation reflected by the Moon during night-time, providing valuable information of atmospheric aerosols during whole day. Therefore, the sun/sky/lunar photometer provides the AOD at night-

time between first and third Moon quarters (Barreto et al., 2013, 2019). The calibration of the CE318-T for AOD calculation at night-time has been done by the Lunar-Langley calibration method explained by Barreto et al. (2019). More details of the sun/sky/lunar photometer Cimel CE318-T and its operational functionalities are described by Barreto et al. (2016).

Furthermore, we used a sky camera SONA ('Sistema de Observación de Nubosidad

Automático': Automatic Cloud Observation System) which provides hemispherical sky images at day and night (González et al., 2012). This system is composed of a CCD camera with a fisheye lens providing RGB images which effective wavelengths at night scenarios correspond to 469, 533 and 608 nm (Román et al., 2017a). It was configured to take multi-exposure sequences of sky images. These sequences are used to obtain a high dynamic range (HDR)

image (one each 5-minutes) which allows, after some correction processes, to obtain the normalized radiances at lunar almucantar points (up to 20° in azimuth from the Moon) at the three effective wavelengths as showed by Román et al. (2017a). A detailed explanation about the configuration, corrections and products obtained by this camera is presented in Román et al. (2017a, b).

**2.3.    In-situ measurements**

The in-situ measurements collected at SNS station are used to assess the aerosol properties, such as scattering coefficient ($\sigma_{sca}$) and volume concentration (VC) retrieved by GRASP algorithm. The integrating nephelometer (model TSI 3563) measures the particle light scattering coefficients at three wavelengths (450, 550 and 700 nm) with 5-min temporal

resolution. A quartz-halogen lamp equipped with a built-in elliptical reflector illuminates over an angle of 7 to 170° the air sample (particle + gas) extracted by a small turbine blower at a constant flow of 30 L min$^{-1}$. The nephelometer measurements underestimate the scattering and backscattering coefficients due to the limits of the angular integration of the scattered light



since a part of forward (0º–7º) and backward (170º–180º) signals are not measured. Nephelometer data have been corrected for truncation and non-Lambertian illumination errors using the method described by Anderson and Ogren (1998).

The Scanning Mobility Particle Sizer (SMPS) composed of an Electrostatic Classifier (TSI Mod. 3080) and a Condensation Particle Counter (CPC; TSI Mod. 3772) measures the sub-micron particle number size distribution within the 12–615 nm particle mobility diameter range with 5-min temporal resolution. Aerosol and sheath flow rates were 1.0 and 5.0 L min$^{-1}$, respectively. SMPS data have been corrected of internal diffusion losses and multiple charges with the AIM software (version 9.0.0, TSI, Inc., St Paul MN, USA). Following calibration procedures, uncertainty in the measured particle size distribution is within 10% and 20% for the size range of 20-200 nm and 200-800 nm respectively (Wiedensohler et al., 2017). In addition, the coarse particle number size distribution within the 0.5–20 µm aerodynamic diameter range was measured by an Aerodynamic Particle Sizer (APS; TSI Mod. APS-3321). The APS measures number aerosol concentrations up to 1000 particles·cm$^{-3}$ with coincidence errors inferior to 5% and 10% at 0.5 and 10 µm diameters, respectively. From these measurements, aerosol volume concentrations were obtained in the 0.05–10 µm radius range with the 5-min time resolution. For that, Q-value=1 has been assumed for conversion from aerodynamic to mobility size distribution (Sorribas et al., 2015).

### 3. Dust event during SLOPE I campaign

This work focuses on an intense dust event that reached the south-eastern of the Iberian Peninsula during SLOPE I field campaign from 18$^{th}$ to 21$^{st}$ July 2016. The analysis of five-day backward-trajectories (Figure 1) computed by HYSPLIT model (Hybrid Single-Particle Lagrangian Integrated Trajectory; Stein et al., 2015; Rolph, 2016) indicate that the air-masses that arrived at Granada came from the southwestern direction on 18$^{th}$ and 21$^{st}$ July 2016. These air-masses from the Sahara Desert area passed along south Morocco and Moroccan coast before reaching Granada. As shown hereafter, these Saharan air masses transported large amounts of Saharan dust particles to the study area.

*[Insert Figure 1 here]*

Figure 2a and b show the day- and night-time AOD at 440 nm (AOD$_{440}$) and the Angrstöm Exponent (AE), computed with AOD at 440 and 870 nm (AE$_{440-870}$), at UGR station provided by AERONET. This figure reveals two different periods throughout the dust event with different AOD$_{440}$ values: the first period from 18$^{th}$ to 19$^{th}$ July (hereinafter P1 period) with





a mean value of 0.50 ± 0.03 and the second period, on 20$^{th}$ to 21$^{st}$ July (hereinafter P2 period) with a mean value (± standard deviation) of 0.94 ± 0.08. These AOD$_{440}$ values obtained in P1 and P2 periods are two and four times higher than the AOD$_{440}$ mean value reported by Perez-Ramirez et al. (2016) for desert dust intrusions over Granada, which make this Saharan dust

event extraordinary. However, the AE$_{440-870}$ values show a smooth behavior with no significant variations around the mean value of 0.17 ± 0.03, which is typical of Saharan dust intrusions over Granada region (e.g. Lyamani et al., 2006; Guerrero-Rascado et al., 2008, 2009; Perez-Ramirez et al., 2016).

*[Insert Figure 2 here]*

Figure 2c and d show the day-to-night temporal evolution of $\sigma_{sca}$ at 550 nm and the total VC obtained from in-situ instrumentation throughout the dust event at SNS high mountain station, respectively. Clearly, both VC and $\sigma_{sca}$ show a continuous increase from the a minimum on 18$^{th}$ July (~50 Mm$^{-1}$ for $\sigma_{sca}$; ~40 μm$^3$/cm$^3$ for VC), up to the maximum values reached early in the morning on 21$^{st}$ July (~350 Mm$^{-1}$ for $\sigma_{sca}$; ~250 μm$^3$/cm$^3$ for VC). This

large increase on these two extensive aerosol properties, especially on 20$^{th}$ - 21$^{st}$ July, is associated with the transport of Saharan dust particles and shows the drastic impact of this Saharan dust event on the aerosol load at SNS remote station.

Figure 3 shows the temporal evolution of the range corrected signal (RCS) at 532 nm from lidar system at UGR station, it points out detailed layers evolution during this Saharan

dust event. The dashed horizontal purple line shows the height of SNS mountain station. This figure reveals important variability in the layer structures of the atmosphere. On 18$^{th}$ July evening two different and decoupled aerosol layers are observed, one at 4 km a.s.l. and the other one near surface up to 1.2 km a.s.l., approximately. However, during the 18$^{th}$ - 19$^{th}$ night the upper aerosol layer went gradually down until it mixed up with the surface aerosol layer

fading away any clear vertical layering. On the following day, particularly after 08:00 UTC, two different aerosol layers are observed again. From the afternoon on 19$^{th}$ to noon on 20$^{th}$, clouds were present over the site and hence the RCS data obtained during this period are excluded from further analysis. After re-starting lidar measurements, on 20$^{th}$ July a decoupled aerosol layer at approximately 4 km a.s.l. is observed again; this layer also went gradually

down until it mixed up with the boundary layer early at night. But the most remarkable observation in this period is the very different structure observed after 04:00 UTC on 21$^{st}$ July when two decoupled layers at ~2-3 km a.s.l. altitude appeared. Later, on 21$^{st}$ July morning, the upper layer collapsed and mixed-up with the surface layer.



*[Insert Figure 3 here]*

The multi-layers aerosol evolution revealed in Figure 3 agree with the observed before in $AOD_{440}$ and $AE_{440-870}$ values and with in-situ measurements at SNS. Actually, the increase in the intensity of RCS on 20[th] July agrees with the increase in AOD. Moreover, the increase of RCS at the altitude of 2500 m a.s.l. coincides with the increase in $\sigma_{sca}$ and VC measured in-situ at SNS station. The collapse of the layer at 2500 m a.s.l. after 08:00 UTC on 21[st] July also agrees with the decrease of $\sigma_{sca}$ and VC at SNS.

Given coherence among all measurements, we can affirm that the Saharan dust affected a wide area and measurements in UGR and SNS are both representative of such event. Thus, the conditions of this dust event allow the evaluation of vertical and columnar aerosol optical and microphysical properties retrieved by GRASP algorithm both day and night-time.

## 4. GRASP retrieval schemes

In this section, we present in four schemes the different strategies used in GRASP algorithm for retrieving continuous day- and night-time atmospheric aerosol properties, both column-integrated and vertical profiles. For daytime retrievals, (denoted as D), the scheme used in GRASP is the proposed by Lopatin et al. (2013) which used as input data both lidar and sun/sky photometer measurements. On the other hand, we have proposed three different schemes to retrieve the aerosol properties during night-time, each scheme can be used depending on the available instrumentation and the conditions of the event.

The lidar data use in each retrieval (both for day- and night-time retrieval) corresponds to preprocessed 30-minute averages of the raw signals for each wavelength. This preprocessing includes background noise subtraction and altitude correction, but other corrections also are applied as overlap correction, analog and photon-counting signals gluing and depolarization correction. To reduce the number of retrieved parameters and to remove the noise in lidar signals at higher altitudes, a logarithmical altitude/range scale with 60 points between a minimum and maximum altitudes is used as in Lopatin et al. (2013). More details of lidar data preprocessing are described in Lopatin et al. (2013). In addition to the lidar signal measurements, each scheme uses different input data from different instrumentation, and hence the retrieval strategies and configurations differ between schemes. These configurations are summarized in Table 1 and described in the following subsections.

*[Insert Table 1 here]*





### 4.1. Daytime scheme

As commented, for scheme D the set of measurements used as input in GRASP are the recommended by Lopatin et al. (2013): the normalized lidar RCS at 355, 532 and 1064 nm and AERONET sun/sky radiances measurements at 440, 675, 870 and 1020 nm. Both daytime AOD and sky radiances used in this work are Version 2 Level 2.0 provided by AERONET from Cimel CE318-T photometer.

### 4.2. Night-time schemes

#### 4.2.1. N0

The first night-time scheme (N0 scheme) used in GRASP assumes that there is no change in the aerosol column integrated extensive and intensive properties along the night. Due to AOD and sky radiance measurements during night-time are still very scarce, this scheme combines the night-time elastic lidar measurements with the closest sun/sky measurements registered the day before or the day after. Thus, the N0 scheme uses the night-time RCS measured by lidar at 355, 532 and 1064 nm combined with the closest daytime measurements of AOD and sky radiances both at 440, 675, 870 and 1020 nm. This scheme offers the possibility to retrieve aerosol vertical properties in stations where night-time photometer measurements are not available, but it only should be applied when the aerosol load and type is similar along night-time.

#### 4.2.2. N1

Currently, night-time AOD measurements, taken with the recently developed sun/sky/lunar photometer CE318-T, are available in some stations. GRASP scheme based on AOD measurements alone was applied by Torres et al. (2017), showing the ability of GRASP to retrieve total column aerosol properties at night-time using this configuration. The present work attempts to go further and provide vertically-resolved aerosol properties at night-time by combining elastic lidar and the night-time AOD at 440, 675, 870 and 1020 nm from lunar photometry measurements.

The second night-time scheme (N1 scheme) approach considers that the aerosol load in the vertical column can be monitored by lunar photometry and hence changes on extensive properties can be detected, but assumes that there are no changes in the aerosol column-integrated intensive properties; therefore, this approach considers that there are no changes in the aerosol type. The N1 scheme proposed in this work combines elastic lidar at 355, 532 and 1604 nm and the night-time AOD at 440, 675, 870 and 1020 nm. As in Torres et al. (2017), the



CRI and the spherical particle fraction are assumed to be known and the values used are the averaged GRASP values retrieved during the closest (after or before) daytime retrievals. AOD at other available wavelengths have been not taken into account in order to choose only the wavelengths used in the scheme D (which is used to extract CRI information at these

wavelengths); in addition night-time AOD values in the UV range is not used due to the low signal in this spectral range at night (Barreto et al., 2019).

### 4.2.3. N2

The third and last night-time scheme (N2 scheme) avoids any assumption as the previous schemes, assuming that intensive and extensive (as N0 scheme) aerosol properties do not

change between day and night, or using a fixed CRI and spherical particle fraction (as N1 scheme). The N2 scheme uses as input data the elastic lidar, lunar aureole normalized sky radiances at 469, 533 and 608 nm derived by the SONA sky camera and the night-time AOD at 440 (which is interpolated to 469 nm by Angström Exponent law using 440 and 675 nm), 675, 870 and 1020 nm. This scheme needs the elastic lidar, lunar photometer and sky camera

measurements but it has the advantage that it is not dependent of daytime measurements and can retrieve extensive and intensive aerosol properties and hence is useful to detect changes on aerosol load or type along the night.

## 5. Results

### 5.1. Columnar aerosol properties

For studying the coherence of daytime columnar-integrated aerosol properties retrieved by GRASP (using D scheme), such retrievals are compared with those provided by AERONET operational algorithm. Generally, the retrievals of Level 2.0 from AERONET Version 2 are used for this comparison, but for specific cases (i.e. $AOD_{440}<0.4$) the SSA and CRI values of Level 1.5 are used instead (Holben et al., 2006). For evaluating columnar aerosol properties

retrieved by GRASP at night-time, we evaluate the smoothness and temporal coherence of the variation of the aerosol retrievals along the night and having as benchmarks the daytime retrievals both AERONET and GRASP D scheme.

The P1 and P2 periods present a situation with an apparent smooth variation of the aerosol load but with the remaining of some intensive properties, identifying the type of

aerosol, along the whole studied period (see Figure 2). In this sense, the selected cases offer an appropriate situation for testing the proposed schemes for night-time aerosol retrievals, having in mind the smoothness of the aerosol evolution in spite of the ample change in the aerosol





load. Hereafter, evaluations of aerosol parameters retrieved by GRASP using different input data set (different schemes) are presented.

### 5.1.1. Columnar particle size distribution parameters

The columnar particle size distribution can be approximated as bimodal log-normals instead of

binned size distributions. The bimodal log-normals can be described using six parameters: volume concentration ($VC_i$ [µm$^3$/µm$^2$]), volume median radius ($r_{v_i}$ [µm]) and standard deviation ($\sigma_{v_i}$) for fine and coarse mode. Table 2 shows the average values (±standard deviation), for all available retrievals, of the size distribution parameters retrieved by GRASP using different configuration schemes and those provided by AERONET. Figure 4 shows the

aerosol size distributions calculated from the parameters given in Table 2. Due to the drastic change in aerosol load (as indicated by AODs) between P1 and P2 periods, the results of GRASP and AERONET retrievals are provided separately for these two periods.

*[Insert Table 2 here]*

*[Insert Figure 4 here]*

15         The aerosol size distribution parameters obtained using scheme D are consistent with AERONET products, with mean relative differences between GRASP and AERONET around 8% (26%), 12% (35%) and 8% (10%) for $VC_c$ ($VC_f$), $r_{v_c}$ ($r_{v_f}$) and $\sigma_{v_c}$ ($\sigma_{v_f}$), respectively, being the agreement better for the coarse mode. In general, the coarse mode parameters obtained during the Saharan dust event analysed here are the typical values obtained at Granada during

dust events originating from Western Sahara (Valenzuela et al., 2012).It is noted that the coarse modal radius retrieved by GRASP D scheme is slightly larger than that provided by AERONET during both periods. This shift towards larger radii for GRASP retrievals was also observed by Lopatin et al. (2013) during dust and biomass burning events over Minsk (Belarus) and by Bovchaliuk et al. (2016) during dust events over Dakar (Senegal) and it is attributed to the use

of additional lidar data.

     Columnar aerosol size distribution parameters at night-time retrieved by GRASP using different schemes (see Table 2) show a good coherence and smooth variation when they are compared against daytime AERONET and GRASP retrievals (scheme D). In fact, the GRASP night-time retrievals using the N0 scheme present average values similar to those provided by

GRASP daytime retrievals with discrepancies around 10% for both modes in the two analysed periods. The aerosol size distribution parameters of coarse mode retrieved by GRASP using N1 scheme are slightly higher systematically than those obtained during day time (by both D





scheme and AERONET) with differences around 15% and 10% for VC$_c$ and r$_{v_c}$, respectively. These differences are inside the uncertainties observed by Torres et al. (2017) in the cases in which the coarse mode is predominant. The use of night-time AOD measurements in N1 scheme, which reveals a change in AOD values (aerosol load) between day and night, can also

be behind these changes in the aerosol size distribution parameters retrieved by N1 scheme.

Finally, the values of aerosol parameters retrieved by GRASP using the N2 scheme are almost similar to the values retrieved by AERONET the day before and after, especially for coarse mode where the discrepancies are around 12%, 3% and 20% for VC$_c$, $r_{v_c}$ and $\sigma_{v_c}$, respectively, showing the potential of such retrievals. However, for fine mode properties (VC$_f$,

$r_{v_f}$ and $\sigma_{v_f}$) there are considerable differences between GRASP and AERONET retrievals mainly due to the low concentration of fine particles.

### 5.1.2. Columnar complex refractive indices

The real (RRI) and imaginary (IRI) refractive indices obtained by GRASP and AERONET are not directly comparable because the GRASP configurations used here provide RRI and IRI

separately for fine and coarse modes while AERONET provides only RRI and IRI equivalent values for the whole size distribution. Nevertheless, the RRI and IRI values provided by AERONET are again used to study the consistency of the proposed schemes for GRASP retrievals. In this case, the mean RRI and IRI values (see Table 3) and their corresponding standard deviations correspond to the whole analysed period. This is done because, in contrast

to VC retrievals that showed a large change between P1 and P2 periods, RRI and IRI retrieved by GRASP (using different schemes) and AERONET were almost stable and showed a very small variation along the whole analysed period, as indicated by the corresponding standard deviations. As can be seen in this table, standard deviations were within and even below the uncertainties associated with the AERONET retrievals, i. e. $\pm 0.03$ for RRI and ±50% for IRI

(Dubovik et al., 2000). On the other hand, it is important to remember that complex refractive indices values for the N1 scheme are not reported in Table 3 because in this case the average day values retrieved by GRASP during daytime were used as input for this GRASP configuration scheme.

*[Insert Table 3 here]*

RRI values retrieved by both GRASP (using different schemes) and AERONET show no remarkable spectral dependence, with maximum spectral variations of 0.03 which is below the uncertainties of the AERONET method. Also, no notable differences are observed between



the retrieved values (using different GRASP schemes) or between the day and night retrievals. Such coherence again shows the potential of the GRASP retrieval proposed. Moreover, retrieved RRI values agree with those reported in previous studies: using AERONET data, Dubovik et al. (2002) reported a mean RRI value of 1.48 ± 0.05 for desert dust at Cape Verde (Republic of Cabo Verde). Also, using GRASP algorithm, Tsekeri et al. (2017) obtained an RRI value of 1.45 for a desert dust event at Finokalia (Crete, Greece). Nevertheless, the RRI values obtained here are lower than those used for desert dust by several models (RRI = 1.53 for the visible spectral region) (Shettle and Fenn, 1979; WMO, 1983, Koepke et al., 1997). However, the differences between RRI values obtained here for desert dust event and those reported in the literature can be explained by the differences in the chemical composition of dust (e.g., Patterson et al. 1977; Carlson and Benjamin, 1980; Sokolik et al., 1993; Sokolik and Toon, 1999).

For IRI, consistency during the whole analysed period is observed again with smooth variations in the retrieved values. For the fine mode, IRI values retrieved by GRASP (using different schemes) show neutral spectral dependence and the differences between the three schemes (D, N0 and N2) are very small with mean difference values around 0.003. But for coarse mode a spectral behaviour of IRI retrieved by GRASP is observed with similar values to the AERONET retrievals. The observed spectral dependence in IRI is the typical observed for desert dust with higher IRI in the UV region (Patterson et al., 1977; Dubovik et al., 2002; Wagner et al., 2012). The mean IRI values retrieved using D and N0 schemes for coarse mode are almost similar to AERONET retrievals being the differences within the uncertainties (about 50%) associated with IRI provided by AERONET (Dubovik et al., 2000). Although the discrepancy between IRI values retrieved using N2 scheme for coarse mode and those provided by AERONET is high, the IRI values of N2 scheme are consistent with IRI values around 0.008 at 675 nm obtained at night-time during a dust event in Dakar (Senegal) by Bovchaliuk et al. (2016). Considering the success in this issue for daytime IRI retrievals, it can be concluded that accurate AOD and sky measurements combined with lidar measurements are useful for accurately characterizing CRI, and particularly for separating the features of fine and coarse modes as discussed by Dubovik et al. (2000). The approach proposed using additional relative radiance in the lunar aureole is also promising for the retrievals of CRI values. Nevertheless, studying the accuracy of the IRI retrieved using night-time sky cameras require further studies.





### 5.1.3.  Columnar single-scattering albedo

Table 4 shows the averaged values of SSA and their corresponding standard deviations obtained by GRASP (using different schemes) and AERONET during the whole dust event. As for IRI and RRI retrievals, SSA values retrieved by both GRASP and AERONET show very small temporal variation during the whole analysed period, as confirmed by the low standard deviations of the SSA values.

*[Insert Table 4 here]*

SSA retrieved by GRASP and AERONET show a smooth variability between day and night for the total period. Actually, mean differences in SSA values retrieved by GRASP and AERONET are below 0.03, which it is within uncertainty associated to AERONET retrieval for dust aerosol (Dubovik et al., 2000), for all the proposed schemes. Moreover, SSA values retrieved by both GRASP and AERONET present a common and remarkable spectral variability with SSA increasing from values around 0.85 at the UV to values around 0.99 in the near-infrared. Such SSA values and spectral dependence with wavelength is typically found in pure desert dust (Dubovik et al., 2002; Valenzuela et al., 2012). These results are in agreement with the observed for IRI, likely indicating that all proposed schemes can extract some information about aerosol absorption from the measurements used as input and/or from the self-retrieval strategy in the case of N1.

### 5.2. Evaluation of vertical aerosol properties

### 5.2.1.  Aerosol extinction profiles at night-time

The multiwavelength lidar system used in this work has two channels detecting Raman scattering at 387 and 607 nm; the Raman method is applied here to independently obtain the aerosol extinction coefficient at 355 and 532 nm. Aerosol extinction profile at 1064 nm is computed using the backscatter coefficient retrieved from the Klett-Fernald method with a constant LR for the entire profile of 50 sr; which is a representative value of desert dust (Guerrero-Rascado et al., 2009). Because Raman measurements of this lidar system are noisy, the lidar signal is averaged ± 15 min around the GRASP retrieval time to get a high signal-to-noise ratio. Raman lidar profiles are obtained with a vertical resolution of 7.5 m and then they are vertically smoothed. The comparison presented hereafter is made between the GRASP values and the Raman values obtained at the closest chosen 60 log-spaced heights in GRASP.

*[Insert Figure 5 here]*



Figure 5 shows the aerosol extinction profiles for the three cases where we have the three night-time schemes retrievals by GRASP and Raman at 355, 532 and 1064 nm. In general, GRASP profiles show similar behaviour than Raman with a slight shift that could be caused by smoothing applied to Raman profiles. The extinction profiles retrieved by GRASP are inside the uncertainties of Raman technique, around 20%, with the exception of the second case (Figure 5b), where the N2 scheme shows large differences with Raman at 355 and 532 nm. For this case, the N2 GRASP retrieval fits worse with Raman likely since the obtained residual error was higher than the residuals of the other retrievals which presented higher convergence.

In order to quantify the agreement between the retrieved extinction with GRASP and Raman, Figure 6 shows aerosol extinction coefficients at 355, 532 and 1064 nm retrieved by GRASP (N0, N1, and N2 schemes) at night-time versus the values obtained by multiwavelength Raman lidar measurements during the dust event observed over Granada in the period $18^{th}$ - $21^{st}$ July 2016. For all schemes and all wavelengths, $\alpha$ retrieved by GRASP and those obtained by Raman lidar measurements are highly correlated with a determination coefficient ($r^2$) ranging from 0.8 to 0.9. The slopes of the regression lines varied between 0.75 and 1.07, indicating that in general the proposed GRASP schemes underestimate the aerosol extinction coefficient obtained by Raman and Klett-Fernald methods. A statistical overview of the differences between $\alpha$ from GRASP retrievals and from Raman measurements is given in Table 5. Particularly, mean and standard deviation of the differences are given by $\Delta\alpha = \alpha_{GRASP} - \alpha_{Raman}$ and also, the average of the relative absolute differences given by $\Delta\alpha(\%) = 100 \cdot |\alpha_{GRASP} - \alpha_{Raman}| / \alpha_{Raman}$ are shown.

***[Insert Figure 6 here]***

The relative differences at 355 nm between $\alpha$ values retrieved by GRASP and those obtained from Raman lidar measurements are around 23% for the three schemes. The lowest bias at 355 nm between values retrieved by GRASP and those obtained from Raman lidar measurements is found for N0 scheme (1.3 ± 40 Mm$^{-1}$), while the highest absolute bias (20 ± 30 Mm$^{-1}$) is obtained for N2 schemes. However, for 532 nm, the differences between GRASP and Raman lidar values are larger than those encountered at 355 nm and 1064 nm, being the relative differences of 30%, 30% and 40% for N0, N1, and N2 schemes, respectively. In addition, the mean biases are higher in this case, being of -30 ± 30 Mm$^{-1}$, -30 ± 30 Mm$^{-1}$ and -40 ± 40 Mm$^{-1}$ for N0, N1, and N2 schemes, respectively. Finally, for 1064 nm, the lowest differences appear for N2 scheme, in opposite with the other wavelengths; the relative differences range from between 20% to 24% for this wavelength. In general, the obtained

GRASP aerosol extinction underestimates Raman measurements at 532 nm for all night-time schemes, while these schemes overestimate the aerosol extinction obtained from Raman measurements at 1064 nm.

*[Insert Table 5 here]*

5       Part of the observed differences could be associated to the assumption associated with the incomplete overlap region, where aerosol properties have been assumed as constant in all this area (Herreras et al., 2019). Also, the rather broad assumption of constant lidar ratio used in the estimation of the extinction at 1064 nm, derived from the backscatter coefficient retrieved by Klett-Fernald retrieval, could explain a part of the observed discrepancies at 1064 nm.

### 5.2.2. GRASP retrievals versus in-situ measurements

Hereafter, $\sigma_{sca}(\lambda)$ and VC retrieved by GRASP are compared versus in-situ measurements obtained at SNS station (2.5 km a.s.l). In Figure 7, the averaged profiles of scattering coefficient at 532 nm (Fig. 7a) and volume concentration profiles (Fig. 7b) retrieved by GRASP night-time schemes are shown. The selected profiles correspond to cases where we have retrievals of all by N0, N1 and N2 schemes, three cases for P1 and six cases for P2 period. In the same figure, we added the averaged in-situ measurements at SNS, both scattering coefficient at 550 nm and volume concentration. N0 and N1 profiles are very similar with low differences (<5%), while N2 profiles have lower values in comparison of these two schemes, with differences around 15%. In the case of low aerosol load (P1 period), GRASP N0 and N1 profiles are closer than in-situ measurements for both $\sigma_{sca}$ and VC, while N2 scheme underestimates these measurements. In contrast, for P2 period, the three schemes show good coherence with in-situ measurements, especially, the VC retrieved by GRASP N2 scheme.

*[Insert Figure 7 here]*

For a direct comparison between GRASP and in-situ measurements we used the averaged values of GRASP retrievals at an altitude of 2.5 ± 0.2 km a.s.l. and in-situ measurements averaged ± 15 min around the GRASP retrieval time. Comparisons of $\sigma_{sca}$ are made at 450, 550 and 700 nm and the AE computed from GRASP retrievals is used to get the equivalent $\sigma_{sca}$ at these wavelengths. Figure 8a shows the temporal evolutions of $\sigma_{sca}$ at 550 nm obtained by GRASP (D, N0, N1, and N2 schemes) and by the integrating nephelometer at SNS station for the analysed dust event. Generally, both GRASP and in-situ measurements follow the same pattern and are sensitive to the arrival of Saharan dust particles. Furthermore, differences between GRASP (using different schemes) and in-situ measurements are very



small, being the differences less than 25 Mm⁻¹ in 90% of the cases. Generally, the differences are negligible for daytime. For night-time, the best agreement is found for the N1 scheme and the worst accordance is obtained for the N2 scheme. The worst accordance for N2 scheme could be due to the smaller scattering angle range of the almucantar radiance retrieved from the moon aureole. In addition, the number of available retrievals for each scheme can be also appreciated in Figure 7; when a retrieval does not appear in the figure is because this retrieval did not pass the imposed convergence criteria.

*[Insert Figure 8 here]*

Figure 9a, b, and c show the scattering coefficients at 450, 550 and 700 nm retrieved by GRASP using the different schemes versus those measured in-situ at SNS. As can be seen in these figures, the measured and the retrieved values are well correlated showing high $r^2$ values between 0.87 and 0.97. Although in general linear fits indicate that GRASP underestimates the in-situ scattering coefficient measurements for low values while shows an overestimation for high values.

*[Insert Figure 9 here]*

An overview of the statistical analysis of the differences between GRASP retrievals and in-situ scattering coefficient measurements is given in Table 6 that shows the mean of the differences expressed as $(\Delta\sigma_{sca} = \sigma_{sca}^{GRASP} - \sigma_{sca}^{SNS})$ and also the mean of the relative differences $\Delta\sigma_{sca} = 100 \cdot |\sigma_{sca}^{GRASP} - \sigma_{sca}^{SNS}|/\sigma_{sca}^{SNS}$ for each scheme. Due to the drastic change in the scattering coefficient between P1 and P2 periods, this statistical analysis is provided separately for these two periods. For the P1 period, GRASP algorithm underestimates the in-situ scattering coefficient measurements both at day and night, especially for N0 and N2 schemes, and at all wavelengths. The highest differences are found for N2 scheme with differences between 30% (at 700 nm) and 35% (at 550 nm). However, for the other schemes (D, N0, and N1) the differences are less than 20%. Again, the uncertainties associated with IRI and with the incomplete overlap assumption as well as the particles losses in sampling inlet can be behind these differences. However, for P2 period, the differences are considerably small and even in some cases they go down to the half of the differences observed in P1 period. On contrary to P1, GRASP overestimates in-situ scattering coefficient in P2 for all schemes except N2. N1, followed by N0, presents the scattering values fitting best with in-situ measurements during P2 period, while D scheme shows the highest differences. The uncertainties associated with IRI and with the incomplete overlap assumption as well as the particles losses in sampling





inlet and also uncertainties in the measurements (used as input in GRASP but also from nephelometer) could be behind at least part of the observed differences.

*[Insert Table 6 here]*

Figure 8b shows the temporal evolutions of the VC retrieved by GRASP at 2500 m a.s.l. and those measured at SNS station. As for the scattering coefficient, the VC retrieved by GRASP and the measured at SNS follow the same pattern both being sensitive to the increase of dust event intensity. Differences at daytime are negligible, while at night-time the differences depend on the GRASP scheme used, being the differences in the P1 period much smaller than in P2 period indicating that the differences increase with increasing aerosol load. Figure 9d shows the VC values retrieved by GRASP (using different schemes) versus those measured at SNS station. The correlation between the measured and the retrieved values is very high with $r^2$ between 0.75 and 0.98. As in the case of the scattering coefficient, linear fits indicate an underestimation by GRASP for low values and overestimates for high values.

*[Insert Figure 10 here]*

Table 7 presents an overview of the statistical analysis of VC comparisons. This table shows the mean of $\Delta VC = VC_{GRASP} - VC_{SNS}$ and the mean of the absolute relative differences described by $\Delta VC(\%) = 100 \cdot |VC_{GRASP} - VC_{SNS}|/VC_{SNS}$. It is clearly observed that GRASP fits the measured values within 15% for D, N0 and N1 schemes during P1 period, while for N2 scheme is observed an underestimation around 30%. However, for P2 period, VC from GRASP overestimates the in-situ measurements with differences around 20% for D, N0 and N1 schemes; while for N2 scheme, GRASP still underestimates the in-situ measurements again but with lower differences, around 10%, than P1 period. The different assumption in GRASP algorithm and the particles losses in sampling inlet (which increase with increasing aerosol load) can be behind the observed differences between GRASP retrievals and in-situ measurements.

*[Insert Table 7 here]*

## 6. Summary and conclusions

The main goal of this work has been to explore the capacity and possibilities of GRASP algorithm to retrieve vertical profiles and column-integrated optical and microphysical aerosol properties at night-time. To this end, we proposed three different schemes combining the measurements of different remote sensing instruments such as elastic lidar, sun/sky/lunar





photometer and/or sky camera. The experimental measurements used in this wok were acquired during a Saharan dust event that took place during SLOPE I campaign at Granada (Spain) from 18$^{th}$ to 21$^{st}$ July 2016. This event has been selected because intensive aerosol properties such as Angström Exponent did not vary too much, with a value around 0.2, and was very intense

with aerosol optical depth (AOD) reaching twice the typical values for Saharan dust outbreaks at Granada.

The three schemes proposed to run GRASP for night-time retrievals have different assumptions, as: no day/night variation of aerosol intensive neither extensive (except vertical distribution) properties (N0 scheme); no day/night variation of aerosol intensive properties but

considering changes on extensive aerosol properties (N1 scheme); day/night variation in both intensive and extensive aerosol properties (N2 scheme).

AERONET inversion products have been used to study the coherence of GRASP night-time retrievals and of the continuous day-to-night aerosol evolution. For the parameters derived from columnar aerosol size distributions, all GRASP schemes show coherent values with

AERONET. Similarly happens for complex refractive index (CRI) and single-scattering albedo (SSA), although more variability is observed, particularly for the N2 scheme due likely to the large uncertainties in relative sky radiance measurements at lunar aureole and the higher freedom degrees assumed than in N1 scheme. Nevertheless, we were not able to go further in the evaluation of the accuracy of the GRASP retrieved parameters. Doing so would require a

large synthetic database that out of the scope of the manuscript. Also, it is needed to study the sensitivity of retrievals to errors in the input optical data, which is the objective of future works.

In general, the aerosol extinction from GRASP and Raman measurements agrees quite well, with differences below 30% at 355, 532 and 1064 nm.

The scattering coefficient ($\sigma_{sca}$) and aerosol volume concentration (VC) retrieved by

GRASP (using different schemes) at 2500 m a.s.l. have been evaluated against in-situ measurements acquired at Sierra Nevada station during a dust event classified in two periods: moderate (P1) and high (P2) aerosol load. Usually, both GRASP retrievals and in-situ measurements follow the same patterns and are sensitive to the arrival of Saharan dust particles. GRASP N0 and N1 schemes underestimate the in-situ $\sigma_{sca}$ and VC measurements for P1 period

(except for VC from N1 scheme) and overestimate for P2 period with differences between 4% and 23%. On the other hand, GRASP N2 scheme underestimates the in-situ measurement both $\sigma_{sca}$ and VC, with differences around 30% and 10% for P1 and P2 periods, respectively. In





general, N2 show slightly high differences than other schemes, however the best results for VC in P2 are for N2 scheme.

The obtained differences could be likely caused by different factors like: the approximation used to fill the incomplete overlap area; the uncertainties in data used as input (large differences shown in N2 scheme could be due the uncertainties associated with the measurements of relative lunar sky radiance); the self-uncertainties of GRASP algorithm under the followed configurations; but also the uncertainty in the values used as reference (like uncertainties in the in-situ measurements); the lack of overlap between night-time retrievals and AERONET daytime retrievals used as reference; and possible inhomogeneity in the atmosphere and local aerosol sources when the GRASP retrievals are compared with in-situ measurements carried out in the mountain.

The analysis presented here is useful to present three configurations of GRASP algorithm to retrieve night-time column-integrated and vertically-resolved of aerosol properties by combination of different remote sensing instruments. In fact, the proposed N2 scheme allows a stand-alone way to retrieve intensive and extensive aerosol properties at night independent on daytime information, even when this scheme usually present higher differences with the reference values. However, this study is only focus in one aerosol episode which is representative of Saharan dust transport and hence, it is necessary to use a more complete dataset that includes at least different aerosol types. Additional studies are needed in this sense to investigate the accuracy and uncertainty of the retrieved GRASP products obtained with the proposed schemes; in this sense sensitivity tests could be done using synthetic data as reference. Therefore, in future studies, it is planned to developed different sensitivity studies with the proposed schemes. In addition, we could try to study the capabilities of GRASP to work with Raman lidar signals and implement the multi-pixel scenario proposed by Dubovik et al. (2011) to retrieve the aerosol properties at night.

*Data availability.* GRASP inversion algorithm software used in this work is free and publicly available at http://www.grasp-open.com. Lidar, lunar photometer and in situ raw data are available from the authors upon request. Sun/sky photometer data are accessible on the AERONET website (http://aeronet.gsfc.nasa.gov/, last access: 25 July 2019).



*Author contributions.* JABO performed the GRASP retrievals, analysed the data and wrote the manuscript. RR processed the sky camera measurements and helped to perform GRASP retrievals. JACV and HL operated and processed the in-situ measurement. The formal analysis, investigation, writing of the original draft, preparation, review of the writing, and editing were performed by JABO,

RR, DPR, JACV, HL and LAA. POA, AEBV, GdAM and JLGR operated the lidar and helped to calculate Raman profiles. AL, DF, MH, BT and OD provided feedback on the GRASP algorithm. AB and PG processed the night-time measurements of the sun/sky/lunar photometer. The project administration, funding acquisition and design of SLOPE campaign were done by FJOR and LAA. All authors provided comments on the manuscript and helped with paper correction.

*Competing interests.* The authors declare that they have no conflict of interest.

*Acknowledgements.* Jose Antonio Benavent-Oltra is funded by MINECO under predoctoral program FPI (BES-2014-068893) and by the University of Granada through "Plan Propio. Programa 7,

Convocatoria 2018". Andrés Bedoya has been supported by a grant for PhD studies in Colombia, COLCIENCIAS (Doctorado Nacional – 647) associated with the Physics Sciences program at the Universidad Nacional de Colombia, Sede Medellín and the Asociación Universitaria Iberoamericana de Postgrado (AUIP). This work was supported the Marie Skłodowska-Curie Research Innovation and Staff Exchange (RISE) GRASP-ACE (grant agreement no778349), by the Spanish Ministry of

Economy and Competitiveness through projects CGL2016-81092-R, CGL2017-85344-R, CMT2015-66742-R, CGL2017-90884-REDT and FIJCI-2016-30007 (Juan de la Cierva-Incorporacion program). Thanks to the AERONET and the ACTRIS/AERONET-Europe networks for the scientific and technical support. The authors thankfully acknowledge the FEDER program for the instrumentation used in this work and the University of Granada that supported this study through the Excellence Units Program.

Special thanks to the Sierra Nevada National Park for its support. Finally, the authors would like to acknowledge the use of GRASP inversion algorithm software ([http://www.grasp-open.com](http://www.grasp-open.com)) in this work.

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



## Tables

Table 1. Data set used as input in GRASP algorithm for day and night-time retrievals. The aerosol size distribution used in each scheme and the number of the converging retrievals obtained during the first (P1) and second (P2)
5  periods.

| | Measurements and wavelengths | | | CRI and sphericity | SD model | Nº Retrievals | |
|---|---|---|---|---|---|---|---|
| | AOD | Sky radiance | RCS | | | P1 | P2 |
| **D** | 440, 675, 870 and 1020 nm | | 355, 532 and 1064 nm | Retrieved | 25 bins | 10 | 7 |
| **N0** | | | | Retrieved | 25 bins | 8 | 16 |
| **N1** | 440, 675, 870 and 1020 nm | x | | Fixed | Bi-modal lognormal | 9 | 11 |
| **N2** | 469, 675, 870 and 1020 nm | 469, 533 and 608 nm | | Retrieved | 25 bins | 6 | 7 |





Table 2. The average values (±standard deviations) of volume concentration (VC$_f$, VC$_c$ [µm³/µm²]), volume modal radius ($r_{v_f}$, $r_{v_c}$ [µm]) and standard deviation ($\sigma_{v_f}$ and $\sigma_{v_c}$) for fine and coarse modes retrieved by GRASP using different configuration schemes and those provided by AERONET. The retrievals are provided for the first period (P1) and the second period (P2). The subscript 'f' denotes fine mode and 'c' denotes coarse mode.

| P1 | AERONET | D | N0 | N1 | N2 |
|---|---|---|---|---|---|
| $VC_f$ | 0.026 ± 0.007 | 0.023 ± 0.009 | 0.020 ± 0.002 | 0.020 ± 0.003 | 0.024 ± 0.002 |
| $r_{v_f}$ | 0.146 ± 0.017 | 0.20 ± 0.03 | 0.202 ± 0.013 | 0.27 ± 0.04 | 0.193 ± 0.007 |
| $\sigma_{v_f}$ | 0.67 ± 0.03 | 0.62 ± 0.06 | 0.63 ± 0.03 | 0.37 ± 0.10 | 0.552 ± 0.019 |
| $VC_c$ | 0.23 ± 0.04 | 0.25 ± 0.05 | 0.28 ± 0.05 | 0.32 ± 0.02 | 0.27 ± 0.03 |
| $r_{v_c}$ | 1.82 ± 0.08 | 2.06 ± 0.10 | 2.09 ± 0.18 | 2.32 ± 0.03 | 1.87 ± 0.14 |
| $\sigma_{v_c}$ | 0.540 ± 0.018 | 0.58 ± 0.03 | 0.57 ± 0.02 | 0.60 ± 0.03 | 0.63 ± 0.03 |

| P2 | AERONET | D | N0 | N1 | N2 |
|---|---|---|---|---|---|
| $VC_f$ | 0.046 ± 0.013 | 0.045 ± 0.014 | 0.037 ± 0.005 | 0.038 ± 0.002 | 0.031 ± 0.006 |
| $r_{v_f}$ | 0.18 ± 0.04 | 0.19 ± 0.04 | 0.16 ± 0.02 | 0.26 ± 0.08 | 0.220 ± 0.009 |
| $\sigma_{v_f}$ | 0.69 ± 0.06 | 0.70 ± 0.10 | 0.74 ± 0.07 | 0.39 ± 0.13 | 0.56 ± 0.04 |
| $VC_c$ | 0.57 ± 0.07 | 0.60 ± 0.06 | 0.65 ± 0.04 | 0.62 ± 0.09 | 0.53 ± 0.04 |
| $r_{v_c}$ | 1.86 ± 0.09 | 2.00 ± 0.09 | 2.03 ± 0.04 | 2.28 ± 0.02 | 1.93 ± 0.13 |
| $\sigma_{v_c}$ | 0.507 ± 0.013 | 0.50 ± 0.04 | 0.51 ± 0.02 | 0.608 ± 0.008 | 0.617 ± 0.011 |





Table 3. The average values (± standard deviation) of the real (RRI) and imaginary (IRI) refractive indices retrieved by GRASP (D, N0 and N2; both fine and coarse modes) and AERONET (for whole aerosol population) during 18th-21st July 2016.

| RRI | | | | | | | |
|---|---|---|---|---|---|---|---|
| | GRASP | | | | | | AERONET |
| | Fine | | | Coarse | | | |
| λ [nm] | D | N0 | N2 | D | N0 | N2 | |
| 355 | 1.44 ± 0.02 | 1.47 ± 0.03 | 1.46 ± 0.01 | 1.45 ± 0.03 | 1.43 ± 0.03 | 1.43 ± 0.01 | |
| 440 | 1.44 ± 0.02 | 1.47 ± 0.03 | 1.46 ± 0.01 | 1.45 ± 0.03 | 1.43 ± 0.03 | 1.43 ± 0.01 | 1.46 ± 0.03 |
| 532 | 1.44 ± 0.02 | 1.47 ± 0.03 | 1.46 ± 0.01 | 1.45 ± 0.03 | 1.43 ± 0.03 | 1.43 ± 0.01 | |
| 675 | 1.44 ± 0.02 | 1.47 ± 0.03 | 1.47 ± 0.01 | 1.45 ± 0.03 | 1.44 ± 0.03 | 1.42 ± 0.01 | 1.47 ± 0.03 |
| 870 | 1.45 ± 0.02 | 1.48 ± 0.03 | 1.45 ± 0.01 | 1.45 ± 0.03 | 1.43 ± 0.03 | 1.41 ± 0.01 | 1.47 ± 0.03 |
| 1020 | 1.44 ± 0.02 | 1.47 ± 0.03 | 1.45 ± 0.01 | 1.45 ± 0.03 | 1.43 ± 0.03 | 1.41 ± 0.01 | 1.45 ± 0.03 |
| 1064 | 1.44 ± 0.02 | 1.47 ± 0.03 | 1.45 ± 0.01 | 1.45 ± 0.03 | 1.43 ± 0.03 | 1.41 ± 0.01 | |

| IRI (x10$^{-3}$) | | | | | | | |
|---|---|---|---|---|---|---|---|
| | GRASP | | | | | | AERONET |
| | Fine | | | Coarse | | | |
| λ [nm] | D | N0 | N2 | D | N0 | N2 | |
| 355 | 2.8 ± 0.4 | 3.2 ± 0.2 | 3.1 ± 0.1 | 10 ± 4 | 8 ± 5 | 12 ± 6 | |
| 440 | 2.8 ± 0.4 | 3.2 ± 0.2 | 3.1 ± 0.1 | 5.5 ± 2.0 | 5.0 ± 2.0 | 9 ± 3 | 5.2 ± 1.4 |
| 532 | 2.8 ± 0.4 | 3.2 ± 0.2 | 3.1 ± 0.1 | 3.4 ± 1.1 | 3.5 ± 1.1 | 6.1 ± 1.6 | |
| 675 | 2.8 ± 0.4 | 3.2 ± 0.2 | 3.1 ± 0.1 | 2.0 ± 0.6 | 2.2 ± 0.5 | 3.4 ± 0.8 | 1.5 ± 0.5 |
| 870 | 2.8 ± 0.4 | 3.2 ± 0.2 | 3.1 ± 0.1 | 1.2 ± 0.4 | 1.5 ± 0.4 | 2.5 ± 0.7 | 1.2 ± 0.4 |
| 1020 | 2.8 ± 0.4 | 3.2 ± 0.2 | 3.1 ± 0.1 | 0.9 ± 0.4 | 1.2 ± 0.4 | 2.2 ± 0.4 | 1.2 ± 0.4 |
| 1064 | 2.8 ± 0.4 | 3.2 ± 0.2 | 3.1 ± 0.1 | 0.8 ± 0.3 | 1.2 ± 0.4 | 2.1 ± 0.4 | |



Table 4. The average values (± standard deviation) of single-scattering albedo (SSA) retrieved by GRASP (using D, N0 and N2 schemes) and AERONET for the period 18th - 21st July 2016 (*469 nm for N2 scheme).

| | SSA | | | | |
|---|---|---|---|---|---|
| | **GRASP** | | | | **AERONET** |
| **λ [nm]** | **D** | **N0** | **N1** | **N2** | |
| **355** | 0.85 ± 0.02 | 0.85 ± 0.03 | 0.83 ± 0.02 | 0.82 ± 0.05 | |
| **440*** | 0.89 ± 0.02 | 0.88 ± 0.02 | 0.88 ± 0.02 | 0.86 ± 0.03 | 0.89 ± 0.03 |
| **532** | 0.93 ± 0.01 | 0.92 ± 0.02 | 0.92 ± 0.01 | 0.89 ± 0.02 | |
| **675** | 0.96 ± 0.01 | 0.95 ± 0.01 | 0.96 ± 0.01 | 0.94 ± 0.01 | 0.97 ± 0.03 |
| **870** | 0.98 ± 0.01 | 0.97 ± 0.01 | 0.98 ± 0.01 | 0.96 ± 0.01 | 0.98 ± 0.03 |
| **1020** | 0.98 ± 0.01 | 0.98 ± 0.01 | 0.98 ± 0.01 | 0.97 ± 0.01 | 0.98 ± 0.03 |
| **1064** | 0.99 ± 0.01 | 0.98 ± 0.01 | 0.98 ± 0.01 | 0.97 ± 0.01 | |

Table 5. Differences (± standard deviation) between the extinction values retrieved by GRASP (N0, N1 and N2 schemes) and Raman during dust event observed over Granada from 18th to 21st July 2016. The percentage differences are between parenthesis.

| $\Delta\alpha(\lambda)$ [Mm$^{-1}$] | N0 | N1 | N2 |
|---|---|---|---|
| **355 nm** | 1.3 ± 40 (23%) | -11 ± 31 (23%) | -20 ± 30 (24%) |
| **532 nm** | -30 ± 30 (30%) | -30 ± 30 (30%) | -40 ± 40 (40%) |
| **1064 nm** | 15 ± 24 (21%) | 20 ± 23 (24%) | 12 ± 22 (20%) |



Table 6. Differences (± standard deviation) between the scattering values retrieved by GRASP (N0, N1 and N2 schemes) and in-situ measurements at SNS provided for the first period (P1) and the second period (P2).

| $\Delta\sigma_{sca}$ [Mm$^{-1}$] | $\lambda$ [nm] | D | N0 | N1 | N2 |
|---|---|---|---|---|---|
| **P1** | **450** | -5 ± 4 (8%) | -17 ± 10 (14%) | -9 ± 19 (13%) | -40 ± 14 (30%) |
| | **550** | -7 ± 8 (10%) | -20 ± 9 (17%) | -14 ± 15 (13%) | -43 ± 13 (40%) |
| | **700** | -5 ± 11 (12%) | -21 ± 9 (19%) | -21 ± 11 (17%) | -36 ± 14 (30%) |
| **P2** | **450** | 40 ± 60 (21%) | 26 ± 17 (13%) | 9 ± 8 (5%) | -31 ± 16 (14%) |
| | **550** | 30 ± 60 (16%) | 11 ± 13 (7%) | 8 ± 7 (4%) | -32 ± 17 (13%) |
| | **700** | 30 ± 60 (16%) | 1.3 ± 12 (4%) | 6 ± 30 (12%) | -17 ± 19 (9%) |

Table 7. Differences (± standard deviation) between the volume concentration values retrieved by GRASP (N0, N1 and N2 schemes) and in-situ measurements at SNS provided for the first period (P1) and the second period (P2)

| $\Delta$VC [μm$^3$/cm$^3$] | D | N0 | N1 | N2 |
|---|---|---|---|---|
| **P1** | -4 ± 9 (14%) | -5 ± 7 (9%) | 1.6 ± 10 (12%) | -21 ± 14 (30%) |
| **P2** | 30 ± 50 (20%) | 29 ± 12 (20%) | 31 ± 32 (23%) | -9 ± 21 (10%) |



**Figures**

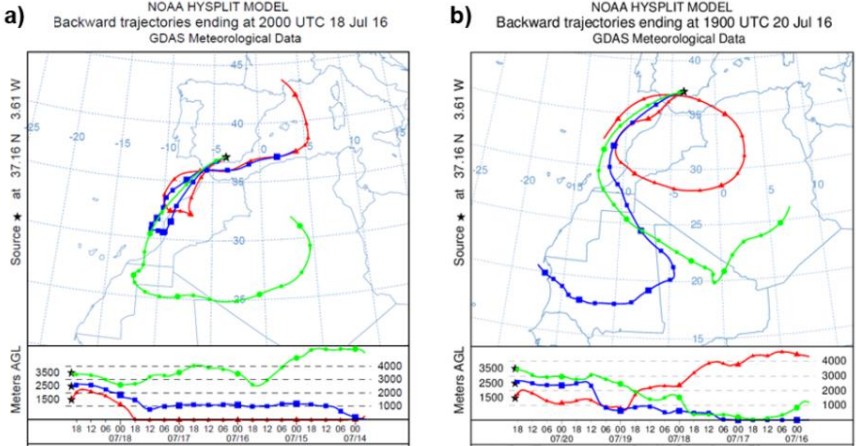

Figure 1. Five-day backward-trajectories computed using HYSPLIT reaching Granada at 20:00 UTC on 18th July (**a**) and at 19:00 UTC on 20th July 2016 (**b**).



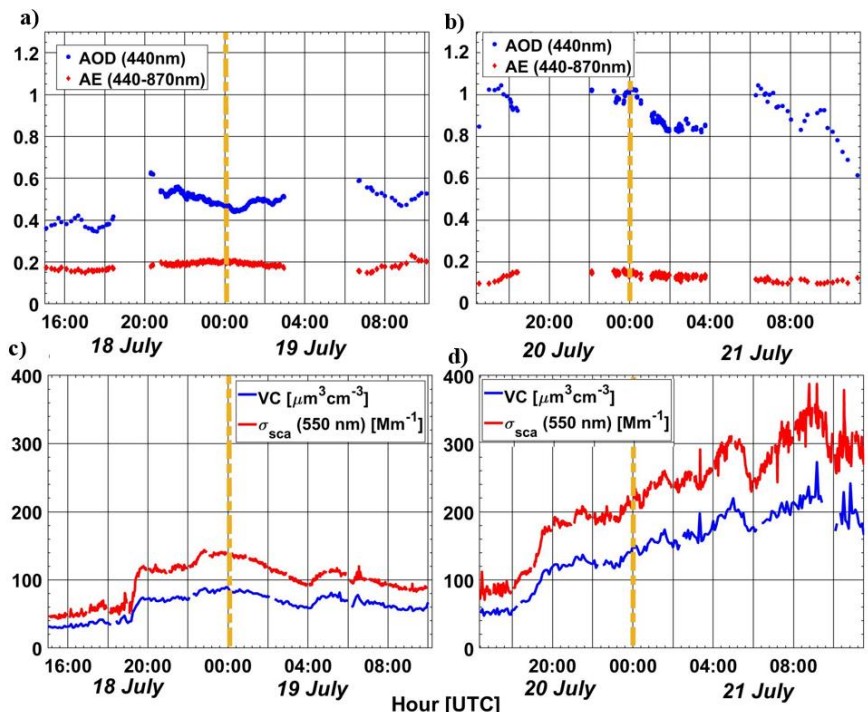

Figure 2. Day-to-night evolution of the AOD at 440 nm (blue) and AE (440-870 nm) (red) obtained at UGR station from 18th to 19th July 2016 (**a**) and from 20th to 21st July 2016 (**b**). Day-to-night temporal evolution of the total volume concentration (VC) and the scattering coefficient ($\sigma_{sca}$) at 550 nm measured at SNS station from 18th to 19th (**c**) and from 20th to 21st (**d**) July 2016.

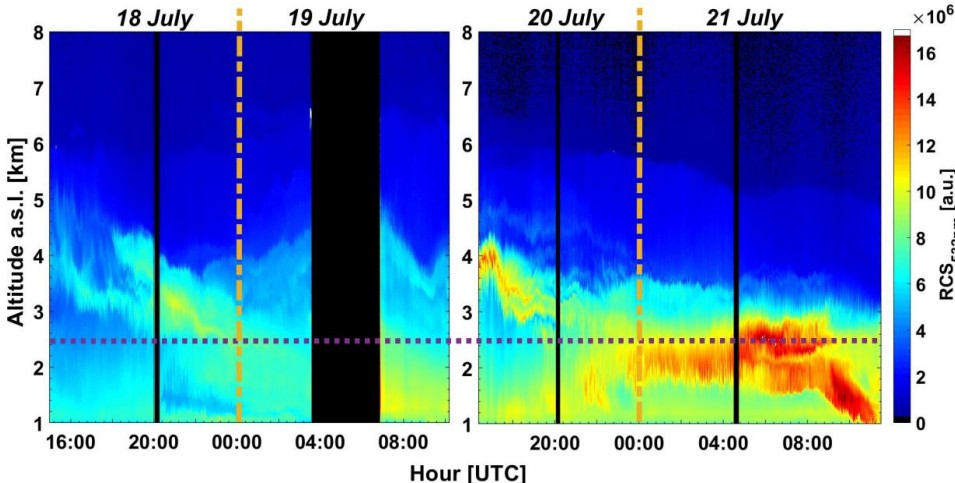

Figure 3. Temporal evolution of the lidar range corrected signal at 532nm from 18th to 19th (**a**) and from 20th to 21st (**b**) July 2016. The purple horizontal line indicates the SNS altitude.





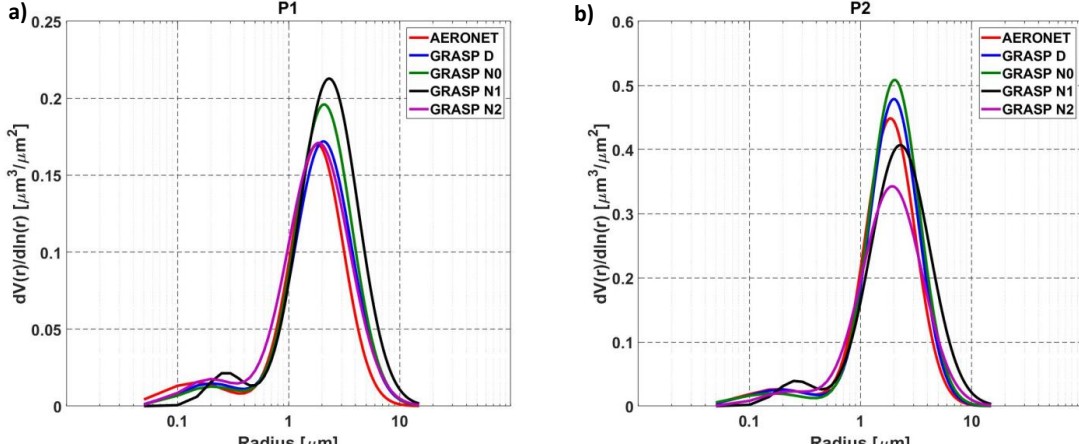

Figure 4. The aerosol size distribution calculated from the averaged aerosol size distribution parameters obtained from AERONET and GRASP retrievals both day and night-time for first period (**a**) and second period (**b**) of dust event.

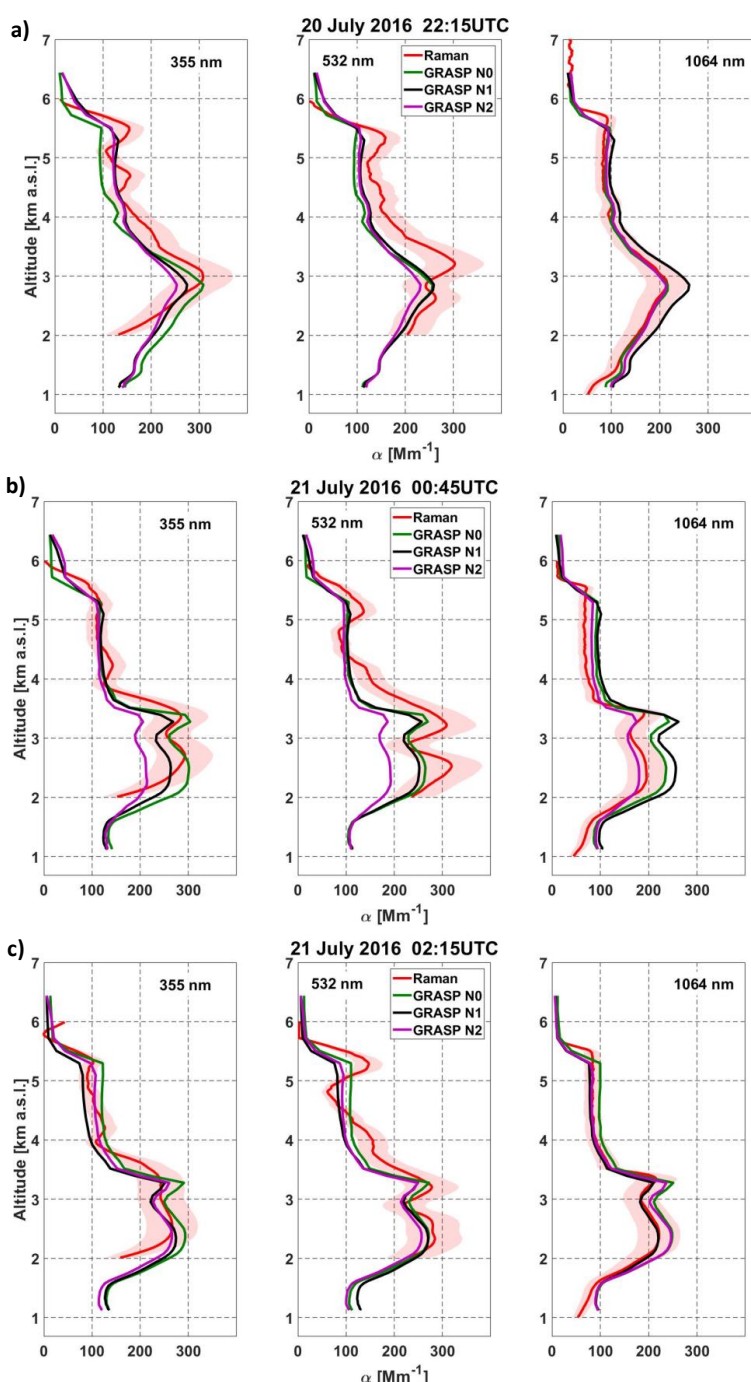

Figure 5. The aerosol extinction profiles retrieved by GRASP (for night-time schemes) and the calculated by
Raman technique at 355 and 532 nm, and Klett-Fernald method at 1064 nm.





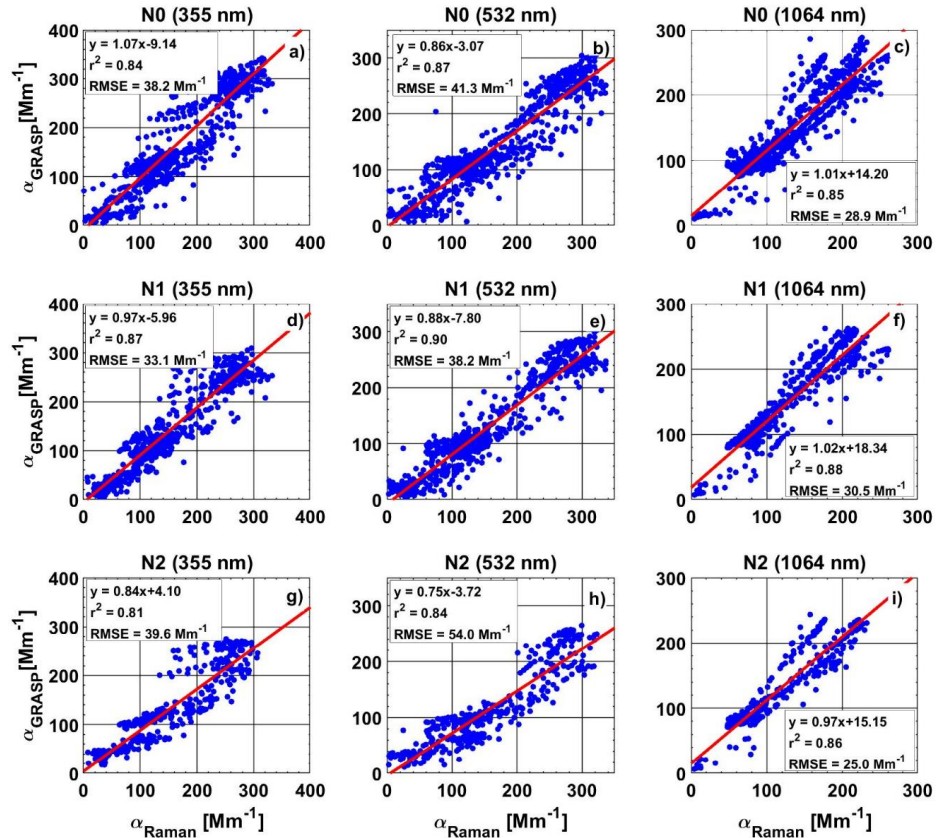

Figure 6. Aerosol extinction coefficient (α) retrieved by GRASP for N0 (**a, b and c**), N1 (**d, e and f**) and N2 (**g, h and i**) as a function of the α calculated by Raman technique at 355 (**a, d and g**) and 532 nm (**b, e and h**) and Klett-Fernald method at 1064 nm (**c, f and i**) during dust event observed over Granada during 18th-21st July 2016.





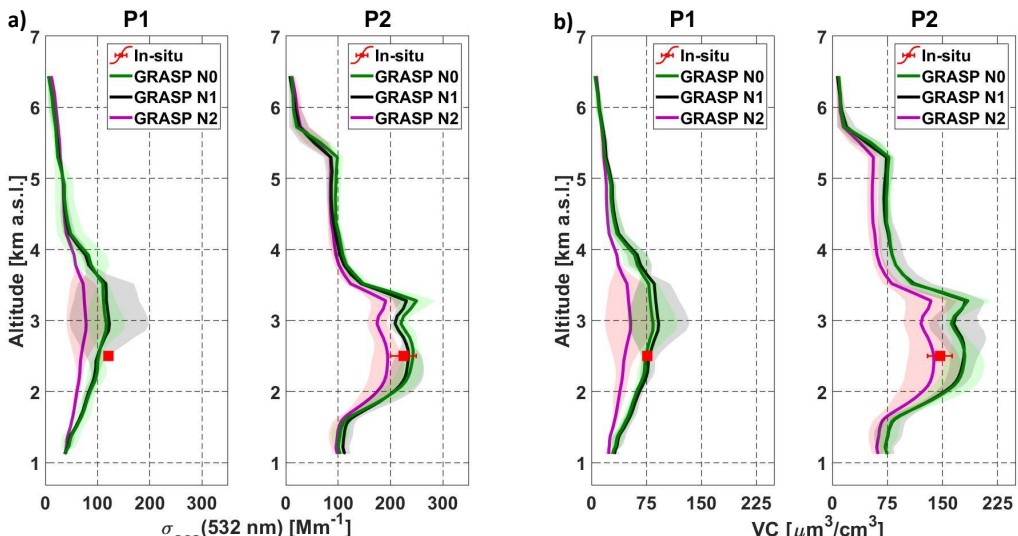

Figure 7. The averaged $\sigma_{sca}$ at 532 nm (**a**) and VC (**b**) profiles with its standard deviation (shaded area) retrieved by GRASP N0, N1 and N2 schemes. In red, the averaged in-situ measurements obtained at 2500 m a.s.l. at SNS station.



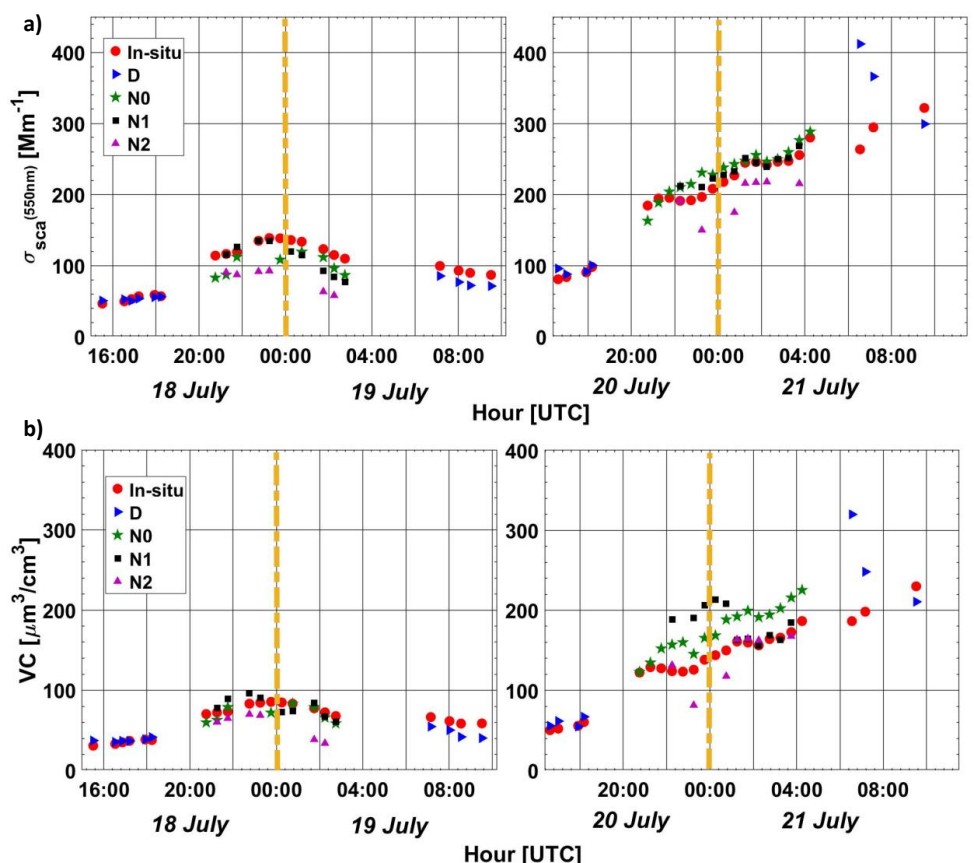

Figure 8. Temporal evolutions of σ$_{sca}$ at 550 nm (**a**) and VC (**b**) measured at Sierra Nevada Station (red) and retrieved by GRASP (D: blue; N0: green; N1: black; N2: cyan) at 2.5 km a.s.l. from 18$^{th}$ to 21$^{st}$ July 2016.





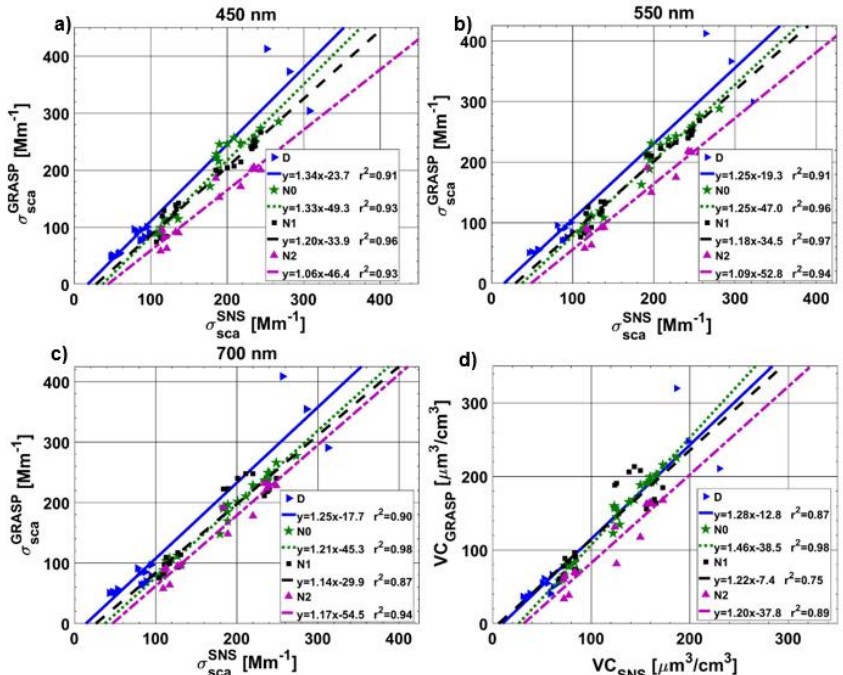

Figure 9. Scattering coefficient, $\sigma_{sca}$, at 450, 550 and 700 nm (**a, b, c**) and volume concentration, VC, (**d**) retrieved by GRASP (D: blue; N0: green; N1: black; N2: purple) at SNS station height versus in-situ scattering coefficient and volume concentration measurements at SNS during dust event over Granada in the period 18th-21st July 2016.

