# Peer review of "Different strategies to retrieve aerosol properties at night-time with GRASP algorithm"

_Atmospheric Chemistry and Physics, 2019_

## Referee Comment (RC1) · Anonymous Referee #1 · 7 Oct 2019

General comments: The manuscript presents the results of a new stage of development and implementation of combined lidar and radiometer sounding (LRS) technique for retrieving aerosol parameters, which extends LRS application to night observations. A distinctive feature of the LRS technique implemented by LIRIC, GARRLiC and GRASP algorithms is that they retrieve the aerosol mode concentration profiles, as well as the complete set of aerosol optical parameters, which determines the heterogeneous aerosol layer in the equations of radiative transfer. Direct and sky radiometric measurements provide information on aerosol extensive and intensive column properties, as well as altitude distributions of aerosol parameters are obtained from multi-wavelength lidar measurements. Radiometric and lidar data form the input data-set for

these algorithms. Manuscript summarizes, proposes and explores a number of strategies for implementation of LRS –technique in the nighttime conditions. The results of this work represent substantial contribution to development of LRS technology on the way to establishment the specific network of combined lidar and radiometer stations.

Specific comments: The authors explore three strategies for retrieving "night-time" aerosol parameters. The results of the measuring campaign in July 2016 present quite convincing confirmation of the applicability of the developed methods for retrieving "night-time" aerosol parameters. However, experimental measurements were performed only for one episode of Saharan dust transport. Additional validation experiments, sensitivity tests and evaluation of uncertainty of the retrieved GRASP products are specified in Sec. 6. as the matter for future study. It is seems, that authors can fine the answer to many questions concerning validation of the GRASP products in the published papers related to LIRIC and GARRLiC because GRASP schemes have a lot in common with these algorithms. 1. Algorism for processing data of D-type (day-time) measurement is identical to GARRLiC. So, the results of sensitivity tests and uncertainty estimations for GARRLiC should be valid for D-strategy. 2. The same results may be used for N0 type of processing. 3. N1-strategy fixes intensive parameters of aerosol modes obtained from of AERONET data. It is LIRIC variant of processing data of LRS measurements. There are many papers aim at studying sensitivity and uncertainty estimations for LIRIC. LIRIC contains a software module for modeling the influence of measurement errors and uncertainties of the aerosol optical parameters specified. 4. Characterization of N2 scheme is a real problem. It is right that "N2 scheme allows a stand-alone way to retrieve intensive and extensive aerosol properties at night independent on daytime information". At the same time, many factors inherent in the sky camera (low sensitivity, high noise and uncertainty of aureole radiance data) are likely to significantly limit the conditions for using the N2 scheme in the nighttime. These restrictions should be defined.

Technical notes: - p. 20: [Insert Figure 10 here] - Figure 10 is absent

Interactive
comment

---

## Referee Comment (RC2) · Anonymous Referee #2 · 10 Oct 2019

The paper "Different strategies to retrieve aerosol properties at night-time with GRASP algorithm" shows novel strategies for the retrieval of vertically-resolved aerosol properties at night-time using GRASP algorithm. To this goal, the authors proposed three different schemes combining the measurements of different remote sensing instruments. To quantify the accuracy of the retrieved night-time aerosol properties obtained by these strategies, the authors used independent aerosol measurements and products as the reference. The title of manuscript clearly reflects the contents of the paper. The paper is well-structured and clearly written. The number and quality of the references are appropriate.

[Figure]

Major comment:

The conclusions in this article are based on experimental data acquired during a Saharan dust event that took place during SLOPE I campaign at Granada (Spain) from 18th to 21st July 2016. Whether they will be valid for another set of experimental data?

Minor comments:

The text of the article contains minor misprints. Namely:

- the punctuation in the first affiliation is not keep;

- the punctuation is not keep in references to the relevant literature, there are no dots or semicolons (for example, P. 2 line 18, P. 6 line 13, P. 4 line 10, P. 15 line 8);

- the numbering is broken for tables 2-3 (P. 31-32);

- figure 10 and its mention are missing from the text (P. 20).

References:

- missing dots at the end of the references (P. 24 line 49, P. 25 line 27, P. 27 line 21, P. 28 lines 20, 22, 24);

- the font size reduced (P. 24 lines 3-8);

- there is no uniform style in writing doi for references.

———————————————————

---

## Author Comment (AC1) · 29 Oct 2019

The paper "Different strategies to retrieve aerosol properties at night-time with GRASP algorithm" shows novel strategies for the retrieval of vertically-resolved aerosol properties at night-time using GRASP algorithm. To this goal, the authors proposed three different schemes combining the measurements of different remote sensing instruments. To quantify the accuracy of the retrieved night-time aerosol properties obtained by these strategies, the authors used independent aerosol measurements and products as the reference. The title of manuscript clearly reflects the contents of the paper. The paper is well-structured and clearly written. The number and quality of the references are appropriate.

We would like to thank the referee #2 for his/her useful comments and suggestions. Responses to the comments are provided next:

*Major comments:*

The conclusions in this article are based on experimental data acquired during a Saharan dust event that took place during SLOPE I campaign at Granada (Spain) from18th to 21st July 2016. Whether they will be valid for another set of experimental data?

In this work we present three different schemes combining the measurements of different remote sensing instruments such as elastic lidar, sun/sky/lunar photometer and/or sky camera to retrieve vertical profiles and column-integrated optical and microphysical aerosol properties at night-time. These schemes are successfully applied to a Saharan dust event, in principle, the results obtained do not indicate that these schemes could not be used for another set of experimental data. However, as stated in the paper, additional studies would be need to investigate the accuracy and uncertainty of the retrieved GRASP products obtained with the proposed schemes, in this sense, sensitivity tests using synthetic data as reference could be done.

*Minor comments:*

The text of the article contains minor misprints. Namely:

- the punctuation in the first affiliation is not keep: Corrected.

- the punctuation is not keep in references to the relevant literature, there are no dots or semicolons (for example, P. 2 line 18, P. 6 line 13, P. 4 line 10, P. 15 line 8): Corrected.

- the numbering is broken for tables 2-3 (P. 31-32): The page 31 and 32 gave the impression that there were two tables in each one, but the Table 2 contains the size distribution parameters both P1 and P2 periods and Table 3 contains the RRI and IRI. The periods have been joined more in Table 2, and the same for RRI and IRI in Table 3.

- figure 10 and its mention are missing from the text (P. 20): Removed "[Insert Figure 10 here]" because this figure not exist.

References:

- missing dots at the end of the references (P. 24 line 49, P. 25 line 27, P. 27 line 21, P.28 lines 20, 22, 24): Corrected.

- the font size reduced (P. 24 lines 3-8): Corrected.

- there is no uniform style in writing doi for references: Corrected.

---

## Author Comment (AC2) · 29 Oct 2019

*General comments:*

The manuscript presents the results of a new stage of development and implementation of combined lidar and radiometer sounding (LRS) technique for retrieving aerosol parameters, which extends LRS application to night observations. A distinctive feature of the LRS technique implemented by LIRIC, GARRLiC and GRASP algorithms is that they retrieve the aerosol mode concentration profiles, as well as the complete set of aerosol optical parameters, which determines the heterogeneous aerosol layer in the equations of radiative transfer. Direct and sky radiometric measurements provide information on aerosol extensive and intensive column properties, as well as altitude distributions of aerosol parameters are obtained from multi-wavelength lidar measurements. Radiometric and lidar data form the input data-set for these algorithms. Manuscript summarizes, proposes and explores a number of strategies for implementation of LRS –technique in the nighttime conditions. The results of this work represent substantial contribution to development of LRS technology on the way to establishment the specific network of combined lidar and radiometer stations.

We would like to thank the referee #2 for his/her useful comments and suggestions. Responses to the specific comments are provided next:

*Specific comments:*

The authors explore three strategies for retrieving "night-time" aerosol parameters. The results of the measuring campaign in July 2016 present quite convincing confirmation of the applicability of the developed methods for retrieving "night-time" aerosol parameters. However, experimental measurements were performed only for one episode of Saharan dust transport. Additional validation experiments, sensitivity tests and evaluation of uncertainty of the retrieved GRASP products are specified in Sec. 6. as the matter for future study. It is seems, that authors can fine the answer to many questions concerning validation of the GRASP products in the published papers related to LIRIC and GARRLiC because GRASP schemes have a lot in common with these algorithms.

1. Algorism for processing data of D-type (day-time) measurement is identical to GARRLiC. So, the results of sensitivity tests and uncertainty estimations for GARRLiC should be valid for D-strategy.

2. The same results may be used for N0 type of processing.

As the referee #1 comments, the input data used in D and N0 schemes are identical to GARRLiC. Thus, these schemes are affected by uncertainties studied in Lopatin et al. (2013) and the quality of their products were evaluated with other algorithms and in-situ measurements in Bovchaliuk et al. (2016), Benavent-Oltra et al. (2017) and Tsekeri et al. (2017). For column-integrated and vertically-resolved aerosol properties, the differences retrieved by GRASP (for D and N0 schemes) with AERONET products or in-situ

measurements are similar to the differences obtained in these previous works. In the next paragraphs we describe the changes done in the manuscript to express these ideas.

We add the following sentence in Section 5.1.2. (page 15, line 14): The differences between GRASP (D and N0 scheme) and AERONET are similar to those obtained in previous works as Benavent-Oltra et al. (2017) and Tsekeri et al. (2017).

We include the following modified sentence in Section 5.1.2. (page 16, line 3): The mean IRI values retrieved using D and N0 schemes for coarse mode are almost similar to AERONET retrievals being the differences within the uncertainties (about 50%) associated with IRI provided by AERONET (Dubovik et al., 2000) and similar to those obtained in previous works as Benavent-Oltra et al. (2017) and Tsekeri et al. (2017).

We include the following modified sentence in Section 5.1.3. (page 16, line 25): Actually, mean differences in SSA values retrieved by GRASP and AERONET are below 0.03, which it is within uncertainty associated to AERONET retrieval for dust aerosol (Dubovik et al., 2000) and similar to those obtained in previous works as Benavent-Oltra et al. (2017) and Tsekeri et al. (2017), for all the proposed schemes.

We add the following sentence in Section 5.2.1. (page 18, line 25): In general, the differences between GRASP and Raman retrievals present in this work are similar to the differences obtained in previous studies (e.g. Bovchaliuk et al., 2016; Benavent-Oltra et al., 2017; Tsekeri et al., 2017).

We add the following sentence in Section 5.2.2. (page 21, line 10): The differences between GRASP (all schemes) and in-situ data are within the differences obtained in previous studies that compared GRASP retrievals with in-situ airborne measurements and LIRIC algorithm (e.g. Bovchaliuk et al., 2016; Benavent-Oltra et al., 2017; Tsekeri et al., 2017).

3. N1-strategy fixes intensive parameters of aerosol modes obtained from of AERONET data. It is LIRIC variant of processing data of LRS measurements. There are many papers aim at studying sensitivity and uncertainty estimations for LIRIC. LIRIC contains a software module for modeling the influence of measurement errors and uncertainties of the aerosol optical parameters specified.

The N1 scheme uses as input data the elastic lidar profiles, AOD night-time, and uses as constraints the aerosol intensive parameters values retrieved using daytime scheme by GRASP (D scheme). This scheme is not exactly the procedure used in LIRIC algorithm that uses as input the lidar data and column-integrated aerosol properties provided by AERONET.

The N1 scheme is based on the approach presented by Torres et al. (2017) that fixes the complex refractive index and sphericity and uses the AOD as input data in GRASP to retrieve the columnar size distribution. In N1 scheme the complex refractive index and sphericity obtained by GRASP in the daytime retrievals (D scheme) are used. The differences between the columnar size distribution parameters obtained using N1 scheme and the daytime retrievals (D scheme) were within the uncertainties described in Torres et al. (2017) (see Sect. 5.1.1).

We add the following sentence in Section 4.2.2. (page 12, line 12): It should be noted that the N1 scheme fixes intensive properties but this scheme is not exactly the same procedure used in LIRIC algorithm that uses as input the lidar data and column-integrated aerosol properties provided by AERONET.

4. Characterization of N2 scheme is a real problem. It is right that "N2 scheme allows a stand-alone way to retrieve intensive and extensive aerosol properties at night independent on daytime information". At the same time, many factors inherent in the sky camera (low sensitivity, high noise and uncertainty of aureole radiance data) are likely to significantly limit the conditions for using the N2 scheme in the nighttime. These restrictions should be defined.

We agree with this comment. To calculate the Moon radiances from sky camera images and avoid the low sensitivity, high noise and uncertainty of aureole radiance data we selected the cases where the scattered light in the lunar aureole region is high. These cases correspond to high values of AOD and high Moon irradiance (at least between the first and last Moon quarters). In addition, we applied some threshold to use the Moon radiance calculated from sky camera images: 1) The Moon zenith angle must be lower than 70º, 2) a minimum of 18 sky radiances with azimuth angles between 3º and 20º must be available for each channel of the sky camera. More details on these procedures can be found in Román et al. (2017a).

To clarify this issue in the paper, we add the following sentence in Section 2.2. (page 7, line 22): Sky cameras usually present a low signal to noise ratio. Thus, to calculate the moon radiances from sky camera images we need cases with high values of AOD (to enhance the scattered moon signal in the aureole) and high Moon extraterrestrial irradiance that restrict data availability to the period between the first and last Moon quarters (Román et al., 2017a). In addition, in this work we applied some threshold to use the Moon radiance calculated from sky camera images: 1) The Moon zenith angle must be lower than 70º, 2) a minimum of 18 sky radiances with azimuth angles between 3º and 20º must be available for each effective wavelengths of the sky camera.

Additionally, we include the following modified sentence in Section 6 (page 23, line 3): In fact, the proposed N2 scheme allows a stand-alone way to retrieve intensive and extensive aerosol properties at night in the cases with high values of AOD and high Moon irradiance (at least between the first and last Moon quarters) independent on daytime information, even when this scheme usually present higher differences with the reference values.

***Technical notes:***

p. 20: [Insert Figure 10 here] - Figure 10 is absent

Removed "[Insert Figure 10 here]" because this figure not exist.